

# Influence of the Pauli principle on two-cluster potential energy

**Yuliya A. Lashko⋆, Victor S. Vasilevsky and Gennady F. Filippov**

Bogolyubov Institute for Theoretical Physics, Kyiv, Ukraine

⋆ ylashko@gmail.com

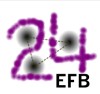

*Proceedings for the 24th edition of European Few Body Conference,
Surrey, UK, 2-6 September 2019*

## Abstract

We study effects of the Pauli principle on the potential energy of two-cluster systems. The object of the investigation is the lightest nuclei of p-shell with a dominant $\alpha$-cluster channel. For this aim we construct matrix elements of two-cluster potential energy between cluster oscillator functions with and without full antisymmetrization. Eigenvalues and eigenfunctions of the potential energy matrix are studied in detail. Eigenfunctions of the potential energy operator are presented in oscillator, coordinate and momentum spaces. We demonstrate that the Pauli principle affects more strongly the eigenfunctions than the eigenvalues of the matrix and leads to the formation of resonance and trapped states.

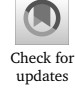
## 1 Introduction

In the present paper, we have considered how the Pauli principle affects a cluster-cluster interaction. We follow a microscopic method, an algebraic version of the resonating group method [1], and consider the lightest nuclei of p-shell with a dominant $\alpha$-cluster channel. Correct account of the Pauli principle is known to be of paramount importance in description of bound and low-energy resonance states in light nuclei with a prominent cluster structure. However, proper account of the antisymmetrization of nucleons belonging to different cluster is quite a tricky problem. Popular cluster models, which treat the antisymmetrization approximately, use a folding cluster-cluster potential. Hence, it is necessary to know what the difference between the folding and exact cluster-cluster potential is. Our paper provides an answer to this question. We found an unexpected impact of the Pauli principle on the potential energy.

Due to the antisymmetrization the potential of interaction between the composite systems is a nonlocal operator. In [2] we formulated an algorithm for studying potential energy of a two-cluster system in a discrete representation. Our method allows reducing the nonlocal interaction to a local form. As a result, one can study effects of the Pauli principle or

effects of other factors or forces on the interaction between complex systems, where the antisymmetrization plays an important role. We would like to stress that this method is quite universal, because it can be applied to any pairs of interacting many-particle systems, such as baryons comprised of quarks, or atoms consisting of electrons.

It is important to notice that the total interaction of two-cluster system originates from nucleon-nucleon interaction and also from the kinetic energy operator. The influence of the Pauli principle on the kinetic energy of relative motion of two clusters has been investigated in Refs. [3–6]. In the present paper we will consider only the first part of the cluster-cluster potential.

The paper is organized as follows. In Section 2 a short explanation of the suggested method is given. Results are discussed in Section 3 and conclusions are stated in Section 4.

## 2 Method

A wave function of $A$-nucleon systems for the partition $A = A_1 + A_2$ is

$$\Psi_{LM} = \widehat{\mathcal{A}} \left\{ [\psi_1 (A_1, s_1, b) \psi_2 (A_2, s_2, b)]_S f_L (q) Y_{LM} (\widehat{\mathbf{q}}) \right\}, \tag{1}$$

where $\psi_\nu (A_\nu, s_\nu, b)$ is a fully antisymmetric function, describing internal structure of the $\nu$th cluster, $\widehat{\mathcal{A}}$ is the antisymmetrization operator permuting nucleons belonging to different clusters and $\mathbf{q}$ is the Jacobi vector determining distance between interacting clusters. We assume that we deal with the s-clusters only, it means that the intrinsic orbital momentum of each cluster equals to zero. The total spin $S$ is a vector sum of the individual spins $s_1$ and $s_2$.

Inter-cluster wave function $f_L (q)$ is a solution to the integro-differential equation. This equation can be much easily solved, when the function (1) is expanded into a complete set of the antisymmetric cluster basis functions

$$|nL\rangle_C = \widehat{\mathcal{A}} \left\{ [\psi_1 (A_1, s_1, b) \psi_2 (A_2, s_2, b)]_S \Phi_{nL} (q, b) Y_{LM} (\widehat{\mathbf{q}}) \right\}, \tag{2}$$

where $n$ is the number of radial quanta, $b$ is the oscillator length $b$. Functions $|nL\rangle_C$ are normalized not to unity, but to eigenvalues $\Lambda_{nL}$ of the norm kernel:

$$\langle nL | \widetilde{n}L \rangle_C = \Lambda_{nL} \delta_{n, \widetilde{n}}.$$

By using the cluster basis functions (2), one obtains the two-cluster Schrödinger equation in the form

$$\sum_{m=0} \left\{ \left\langle nL \left| \widehat{H} \right| mL \right\rangle_C - E \Lambda_{nL} \delta_{n,m} \right\} C_{mL} = 0, \tag{3}$$

where $\left\langle nL \left| \widehat{H} \right| mL \right\rangle_C$ is a matrix element of a microscopic two-cluster Hamiltonian, $C_{nL}$ is the expansion coefficient.

If we omit the antisymmetrization operator in the expression for the wave function, we have got the so-called folding approximation.

$$\Psi_{LM}^{(F)} = [\psi_1 (A_1, s_1, b) \psi_2 (A_2, s_2, b)]_S f_L^{(F)} (q) Y_{LM} (\widehat{\mathbf{q}}). \tag{4}$$

This approximate form is valid when the distance between clusters is large and effects of the Pauli principle are negligible small. The exact two-cluster potential is a nonlocal operator, as opposed to the folding cluster-cluster potential. The idea is to compare the exact and folding two-cluster potentials via separable representation of the potentials. As a tool for this study we employ the method suggested in our recent paper [2]. We will construct matrix of potential

energy and then analyze its eigenvalues and eigenfunctions of the matrix. The eigenfunctions will be analyzed in the oscillator, coordinate and momentum representations. Involving three different spaces allows us to get more complete picture on the nature and properties of potential energy eigenfunctions.

Having constructed matrix of potential energy $\left\| \left\langle nL \left| \widehat{V} \right| mL \right\rangle \right\|$ of dimension $N \times N$, we can calculate eigenvalues $\lambda_\alpha$ ($\alpha=1, 2, \ldots, N$) and corresponding eigenfunctions $\left\{ U_n^\alpha \right\}$ of the matrix. Diagonalization of the potential energy matrix generates a new set of inter-cluster functions $\phi_\alpha$ and two-cluster wave functions $\Psi_\alpha$

$$\phi_\alpha(q, b) \;=\; \sum_n U_n^\alpha \Phi_{nL}(q, b) \tag{5}$$

$$\Psi_\alpha \;=\; \widehat{\mathcal{A}}\left\{ \psi_1(A_1) \psi_2(A_2) \phi_\alpha(q, b) Y_{LM}(\widehat{\mathbf{q}}) \right\}. \tag{6}$$

The functions $\phi_\alpha(q, b)$ and eigenvalues $\lambda_\alpha$ enable us to construct inter-cluster nonlocal potential

$$\widehat{V}_N(q, \widetilde{q}) = \sum_{\alpha=1}^N \phi_\alpha(q, b) \lambda_\alpha \phi_\alpha(\widetilde{q}, b). \tag{7}$$

In what follows we are going to study properties of the eigenvalues and eigenfunctions of the potential energy operator in the oscillator representation $\left\{ U_n^\alpha \right\}$, coordinate $\phi_\alpha(q, b)$ and momentum $\phi_\alpha(p, b)$ spaces.

For a two-body case, the eigenfunctions $\phi_\alpha(q, b)$ or $\phi_\alpha(p, b)$ would immediately define a wave function and t-matrix, as it was demonstrated in Ref. [2]. However, in two-cluster systems the antisymmetrization is known to affect the kinetic energy and norm kernel and thus the kinetic energy and norm kernel participate in creating the effective cluster-cluster interaction as well.

In the present paper we consider only the part of the cluster-cluster potential generated by the nucleon-nucleon potential with the focus on the Pauli effects. The first effect of the Pauli principle on two-cluster systems is connected with appearance of the Pauli forbidden states, which correspond to zero eigenvalues of the norm kernel. The second effect of the Pauli principle is related to the eigenvalues of the Pauli-allowed states which are not equal to unity. It has been shown in [3] that the kinetic energy operator of two-cluster relative motion modified by the Pauli principle generates an effective interaction between clusters. It is interesting to analyze how the eigenvalues of the norm kernel change potential energy of two-cluster system.

## 3 Results and discussion

The object of the investigation is the lightest nuclei of p-shell with a dominant alpha-cluster channel shown in Table 1.

Table 1: List of nuclei and two-cluster configurations

| Nucleus | Configuration |
|---------|---------------|
| $^5$He | $^4$He$+n$ |
| $^5$Li | $^4$He$+p$ |
| $^6$Li | $^4$He$+d$ |
| $^7$Li | $^4$He$+^3$H |
| $^7$Be | $^4$He$+^3$He |
| $^8$Be | $^4$He$+^4$He |

Table 2: Oscillator length $b$ in fm for different nuclei and different potentials.

| Nucleus | VP | MHNP | MP |
|---|---|---|---|
| $^5$He, $^5$Li | 1.38 | 1.32 | 1.28 |
| $^6$Li | 1.46 | 1.36 | 1.31 |
| $^7$Li, $^7$Be | 1.44 | 1.36 | 1.35 |
| $^8$Be | 1.38 | 1.32 | 1.28 |

We employ three nucleon-nucleon potentials which have been often used in different realizations of the cluster model. In our calculations we involve the Volkov N2 (VP) [7], modified Hasegawa-Nagata (MHNP) [8,9] and Minnesota (MP) [10] potentials. Coulomb forces are also involved in calculations and treated exactly. For the sake of simplicity we neglect the spin-orbit forces, thus the total spin $S$ and the total orbital momentum $L$ are good quantum numbers. Oscillator length $b$ is selected to optimize energy of the lowest decay threshold for each nucleus and for each NN potential. In what follows, it is assumed the energy of two-cluster systems is determined with respect to the two-cluster threshold.

The optimal values of $b$ are shown in Table 2.

Figure 1 shows the eigenvalues of the exact and folding potential energy matrix generated by the MHNP for the $1^-$ state of $^7$Be.

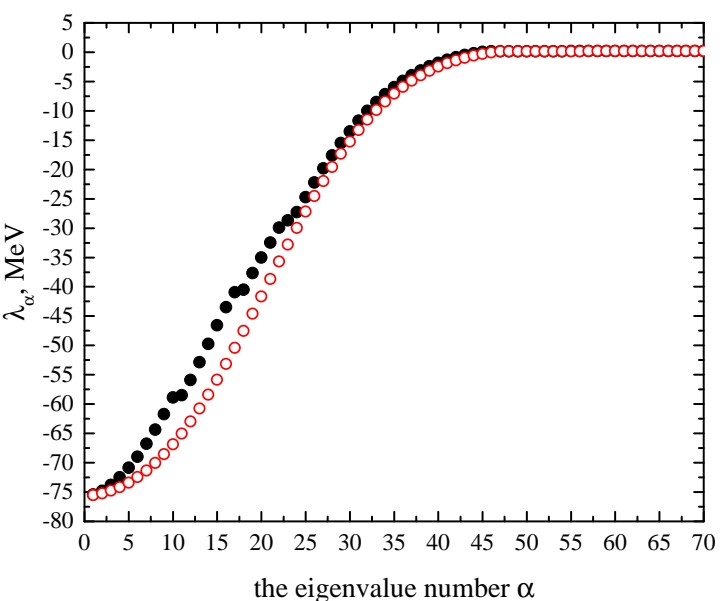

Figure 1: Eigenvalues of the exact (solid circles) and folding (open circles) potential energy matrix for the $1^-$ state of $^7$Be. Results are obtained with the MHNP.

We can observe from Fig. 1 that the eigenvalues of the potential energy matrix calculated with antisymmetrization are very close to those determined in the folding approximation. The lowest eigenvalues almost coincide indicating that both potentials have the same depth. One can also see that exact potential is less attractive at the range $5 < \alpha < 30$. For $\alpha > 50$ the exact potential is very close to the folding potential. Similar behavior of eigenvalues is observed for all lightest nuclei of the p-shell and for all NN potentials involved in our calculations.

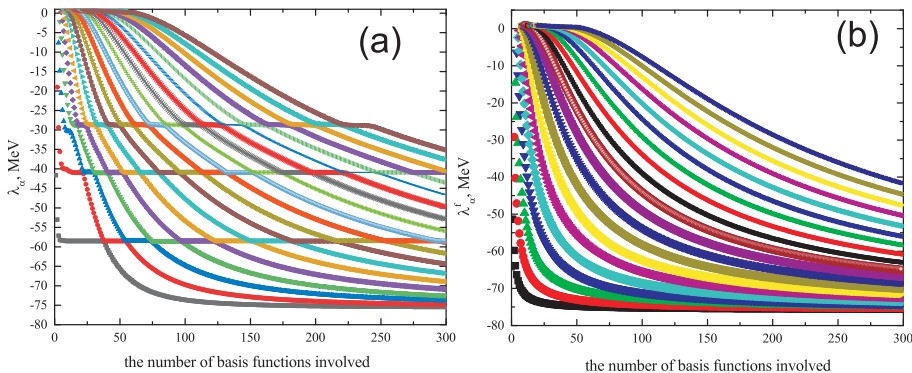

Figure 2: Eigenvalues of the exact (a) and folding (b) potential energy matrix as a function of the number $N$ of oscillator functions involved in calculations. Results are obtained for the $1^-$ state in $^7$Be with the MHNP.

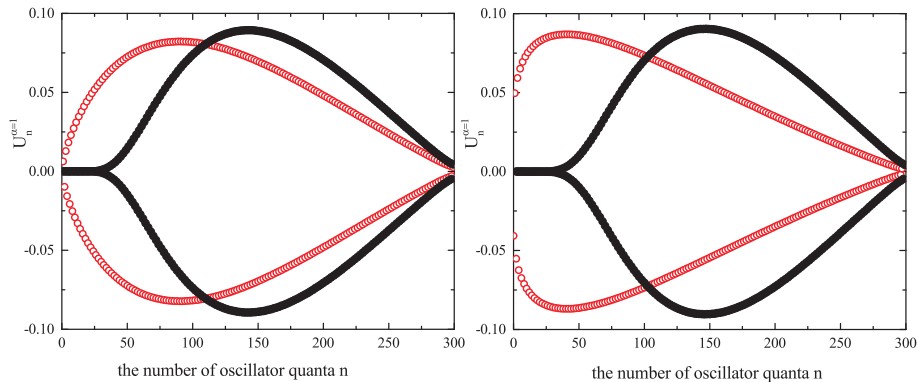

Figure 3: The eigenfunctions of the exact (solid circles) and folding (open circles) potential energy in oscillator representation for the $1^-$ state in $^7$Be (left panel) and the $0^+$ state in $^8$Be (right panel). Results are obtained with the MHNP.

In Fig. 2 we show dependence of the eigenvalues $\lambda_\alpha$ on the number of oscillator functions involved in calculations. These results are obtained for $L^\pi = 1^-$ state of $^7$Be with the MHNP. As can be seen from Fig. 2, the dependence of eigenvalues of the exact potential on the number of functions exhibits resonance behavior. Contrary, none of the eigenstates of the folding potential has a resonance behavior.

In Fig. 3 we compare eigenvectors for $^8$Be and $^7$Be with and without antisymmetrization for the MHNP. One can see that they are quite different. The Pauli principle makes zero the first 50 expansion coefficients $U_n^\alpha$. So, we can conclude that the eigenfunctions of the exact potential corresponding to non-resonance values of $\alpha$ are suppressed at the range $n < 50$ due to the influence of the Pauli principle. The eigenfunctions of the folding potential have a maximum at lower number of quanta than the eigenfunctions of the exact potential. It is also worth noting that different behaviour of the eigenfunctions of the folding potential at small values of $n$ for $^7$Be and $^8$Be is caused by different values of orbital momenta.

Figure 4 presents the eigenfunctions of the potential energy operator for the $0^+$ state in $^8$Be in the momentum space for $\alpha = 1, 2$ and 3. A huge repulsive core in the MHNP and the Pauli principle make eigenfunctions $\phi_\alpha(p)$ to vanish in a large range of $0 < p < 10$ fm$^{-1}$.

Now let us consider wave functions of trapped and resonance states in the two-cluster systems. Wave functions of the resonance states in $^{8,7}$Be and $^6$Li in oscillator and momentum

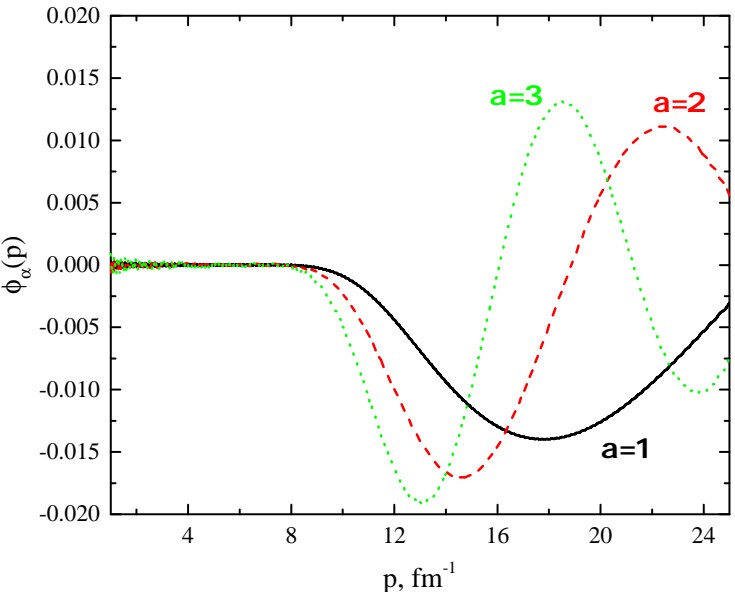

Figure 4: The eigenfunctions of the potential energy operator for the $0^+$ state in $^8$Be in the momentum representation. Results are obtained with the MHNP.

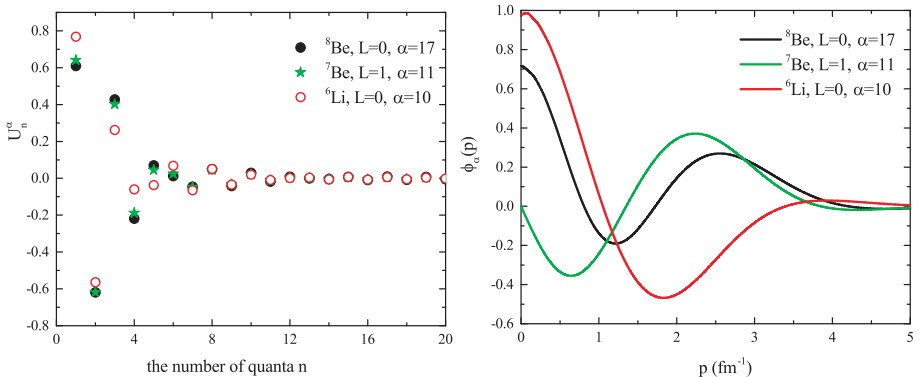

Figure 5: Wave functions of the resonance states in $^{8,7}$Be and $^6$Li in oscillator representation (left panel) and momentum representation (right panel). Results are obtained with the MHNP

representation are shown in Fig. 5. We can conclude that the eigenfunctions of resonance states describe a compact configuration, because they are localized at low values of oscillator quanta and momentum.

Eigenvalues of the potential energy matrix generated by the Volkov N2 potential are shown in Fig. 6 for the $1^-$ state of $^5$He. The eigenvalues of the exact VP differ from those of the folding potential at a single point $\alpha = 1$. The eigenvalue $\lambda_{\alpha=1}$ corresponds to a "trapped" state. This conclusion follows from Fig. 7, where the dependence of the eigenvalues of the potential energy matrix as a function of the number of oscillator functions involved in calculations is shown for the $1^-$ state in $^5$He. Figure 7 shows fast convergence of the first eigenvalue of the exact potential energy matrix.

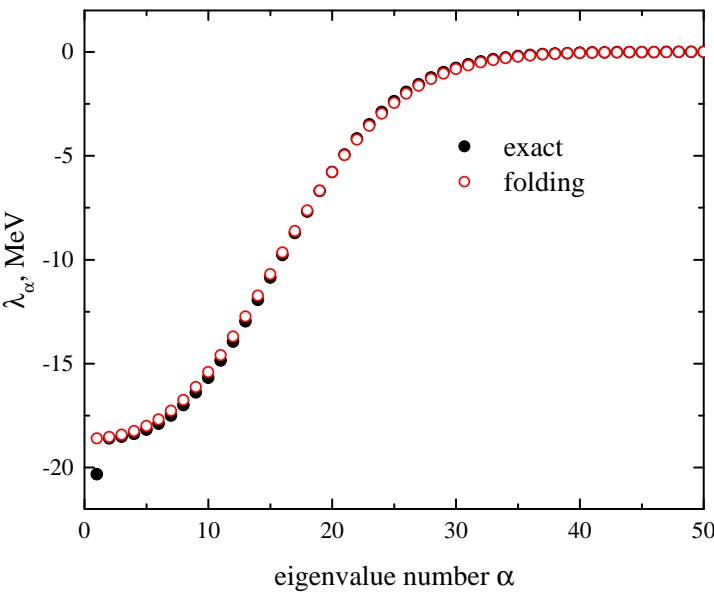

Figure 6: Eigenvalues of the exact (solid circles) and folding (open circles) potential energy matrix for the $1^-$ state of $^5$He. Results are obtained with the VP.

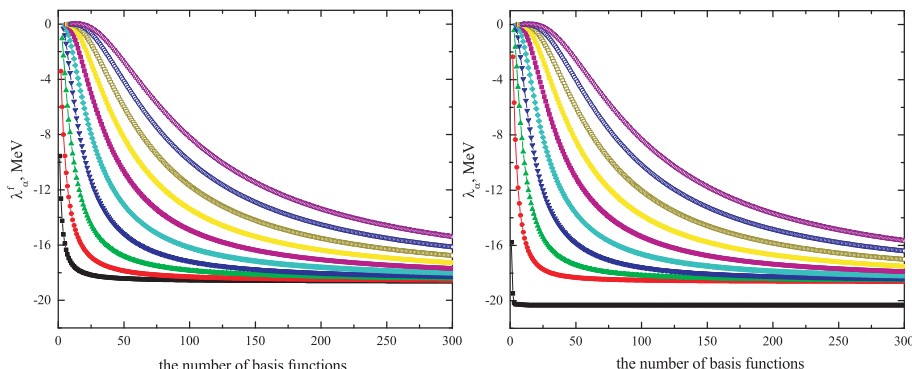

Figure 7: Eigenvalues of the exact (right pane) and folding (left panel) potential energy matrix as a function of the number of oscillator functions involved in calculations. Results are obtained for the $1^-$ state in $^5$He with the VP.

Wave functions of the trapped state in $^5$He, $^6$Li and $^7$Be in oscillator representation are shown in Fig. 8. The wave functions of the trapped state have an exponential asymptotic behavior. Thus, there is a full resemblance of these functions with a true bound state wave function which is usually observed in coordinate space. The asymptotic part of the wave functions of resonance states has an oscillatory behavior (Fig. 5). The node of the trapped state wave functions appears due to the orthogonality of this state to the Pauli forbidden states in two-cluster systems.

It is interesting to note that a "trapped" state in the VP appears only for the cluster configurations characterized by the eigenvalues of the norm kernel $\Lambda_n > 1$. Namely, the VP generates a "trapped" state in the states of normal parity in $^5$He, $^{5,6,7}$Li and $^7$Be. MP also produces a

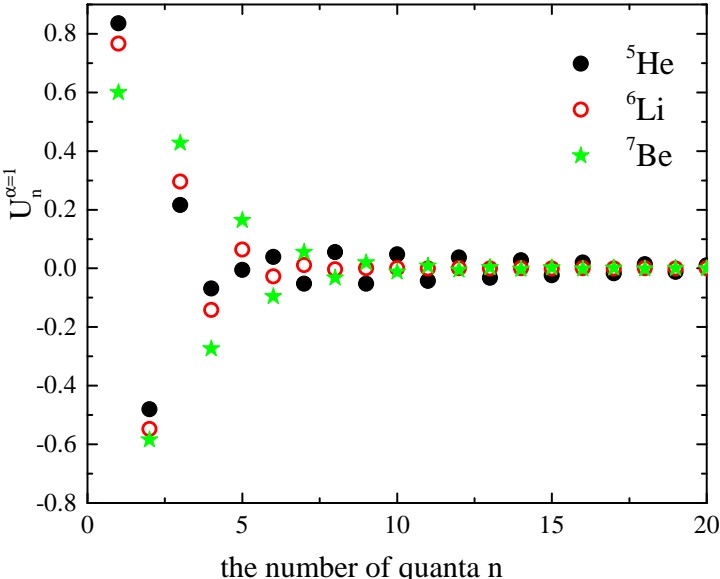

Figure 8: Wave functions of the trapped state in $^5$He, $^6$Li and $^7$Be in oscillator representation. Results are obtained with the VP.

"trapped" state in $^6$Li.

## 4 Conclusion

We have studied influence of the Pauli principle on the interaction between two clusters within a microscopic method – an algebraic version of the resonating group method. Due to the Pauli principle, a cluster-cluster interaction is a nonlocal potential within the standard version of the resonating group method. We employed the algebraic version of the method, which involves a complete basis of oscillator functions to expand a two-cluster wave function. In the framework of the latter method, a nonlocal cluster-cluster interaction was represented as a matrix. Our main aim was to study properties of matrix of the potential energy operator generated by a nucleon-nucleon and Coulomb potentials. We constructed the matrix with and without full antisymmetrization. These two matrixes allowed us to reveal explicitly the influence of the Pauli principle on the shape of the cluster-cluster interaction.

Eigenvalues and eigenfunctions of the folding cluster-cluster potential have been compared with those of the non-local cluster-cluster potential for the lightest nuclei of the $p$-shell: $^5$He, $^5$Li, $^6$Li, $^7$Li, $^7$Be and $^8$Be. All these nuclei were considered as two-cluster systems composed of an alpha particle and a nucleus of the $s$-shell. We employed the Minnesota potential, the modified Hasegawa-Nagata potential and Volkov N2 potential to investigate the dependence of cluster-cluster interaction on the shape of the potential.

It was demonstrated that eigenvalues of the folding two-cluster potential coincide with the potential energy in coordinate space at some specific discrete points. It was also shown that the eigenfunctions of the folding potential energy matrix are the expansion coefficients of the spherical Bessel functions in a harmonic oscillator basis.

In general, the eigenvalues of the folding and exact cluster-cluster potential do not diverge

considerably. However, the dependence of the exact cluster-cluster potential on the number of the invoked functions reveals a number of resonance states which are absent in the case of folding potential. The structure of the resonance states is much different from the eigenfunctions of the folding potential. Such resonance states are mainly localized in the region of small number of quanta in discrete space and, consequently, in the region of small distances between clusters in coordinate space.

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
