# Peer review of "Influence of the Pauli principle on two-cluster potential energy"

_SciPost Physics Proceedings, doi:SciPost Phys. Proc. 3, 021 (2020)_

## Round 1 · Referee Report · Anonymous (Referee 1) · 2019-11-18

Strengths

1) Authors demonstrated that for the non-compact cluster shapes the eigenvalues of the exact and folding potential energy matrix coincide.

2) For the compact cluster shapes with small relative distances between clusters, authors found a number resonance states with the exact cluster-cluster potential. In contrast, these resonance states are absent in the case of folding potential.

Weaknesses

No weakness

Report

In the present manuscript, the influence of the Pauli principle on the cluster-cluster potential energy of the two-cluster nuclear system was studied in the algebraic version of the resonating-group method. Authors used three well-known nucleon-nucleon potentials: Volkov N2, modified Hasegawa-Nagata, and Minnesota potentials. A few interesting results were obtained by authors. For the lightest nuclei of $p$-shell, $^{5}$He, $^{5,6,7}$Li and $^{7,8}$Be considered as two-cluster systems, the eigenvalues and eigenfunctions of the exact (with full anti-symmetrization) and folding (without full anti-symmetrization)
potential energy matrix were compared. Authors demonstrated that for the non-compact cluster shapes the eigenvalues of the exact and folding potential energy matrix coincide. For the compact cluster shapes with small relative distances between clusters, the exact cluster-cluster potential shows a number resonance states. In contrast, these resonance states are absent in the case of folding potential.

So, there is no doubt that the subject studied in the manuscript is very important and actual for the nuclear cluster community to justify publication.

In total, the manuscript is written very well and cleanly. There are no typos. The reference list is complete. Therefore, the manuscript certainly deserves a publication.

Requested changes

No changes

---

## Editorial Decision

published